# Deep probabilistic 3D angular regression for directional dark matter detectors

## Abstract

Modern detectors of elementary particles are approaching a fundamental sensitivity limit where individual quanta of charge can be localized and counted in 3D. This enables novel detectors capable of unambiguously demonstrating the particle nature of dark matter by inferring the 3D directions of elementary particles from complex point cloud data. The most complex scenario involves inferring the initial directions of low-energy electrons from their tortuous trajectories. To address this problem we develop and demonstrate the first probabilistic deep learning model that predicts 3D directions using a heteroscedastic von Mises-Fisher distribution that allows us to model data uncertainty. Our approach generalizes the cosine distance loss which is a special case of our loss function in which the uncertainty is assumed to be uniform across samples. We utilize a sparse 3D convolutional neural network architecture and develop approximations to the negative log-likelihood loss which stabilize training. On a simulated Monte Carlo test set, our end-to-end deep learning approach achieves a mean cosine distance of $0.104\ (26°)$ compared to $0.556\ (64°)$ achieved by a non-machine learning algorithm. We demonstrate that the model is well-calibrated and allows selecting low-uncertainty samples to improve accuracy. This advancement in probabilistic 3D directional learning could significantly contribute to directional dark matter detection.

## 1 Introduction

Elusive, electrically neutral elementary particles can be detected by measuring the recoil trajectories of atomic nuclei or electrons produced when the neutral particles scatter in a detector. Modern gaseous detectors are uniquely capable of reconstructing the directions of such recoils. This enables a highly desirable capability: directional detection of dark matter, neutrinos, and neutrons.

The benefits of obtaining directional information are profound and could prove game-changing in several areas of fundamental physics. Most importantly, dark matter detection experiments could reject solar neutrino backgrounds based on recoils pointing back towards the sun, and positively identify dark matter from the observed angular distribution of recoils (Billard et al., 2014; O'Hare, 2021). This would overcome two major limitations in current experiments seeking to identify the particle nature of dark matter, which is arguably the most urgent and exciting problem in contemporary physics. Directional measurements of electron recoils induced by solar neutrinos would enable improved measurements of neutrinos from the Sun's "CNO cycle" which could settle a long-standing puzzle known as the solar abundance problem (Villante & Serenelli, 2019; O'Hare et al., 2022). Directional measurements of photoelectrons produced by X-rays enable us to map the X-ray polarization of extended astrophysical sources, as in the Imaging X-ray Polarimetry Explorer (Weisskopf et al., 2016). A comprehensive review of directional recoil detection and its benefits can be found in Vahsen et al. (2021).

The gas time projection chamber (TPC) is the most mature directional recoil detection technology. Gas TPCs reconstruct the ionization track of the recoiling nucleus or electron. An example of an electron recoil detected in 3D using a gas TPC developed by Jaegle et al. (2019) is shown in Figure 1. The detector provides a voxel grid where each voxel is given a value corresponding to the amount of ionization detected at its position. Inferring the initial direction of the recoiling particle from this data is a difficult challenge. There are three main obstacles: First, the electron track is not straight, as the electron scatters in the gas. Second, the recoil starts and ends inside the detector, and it is

not clear which side is the beginning, and which side is the end. Third, the non-straight trajectory and the start-versus-end ambiguity both make it difficult to estimate the uncertainty of any estimated direction. Unlike many common computer vision problems, even a trained human expert, such as a graduate student, would have difficulties correctly interpreting this data.

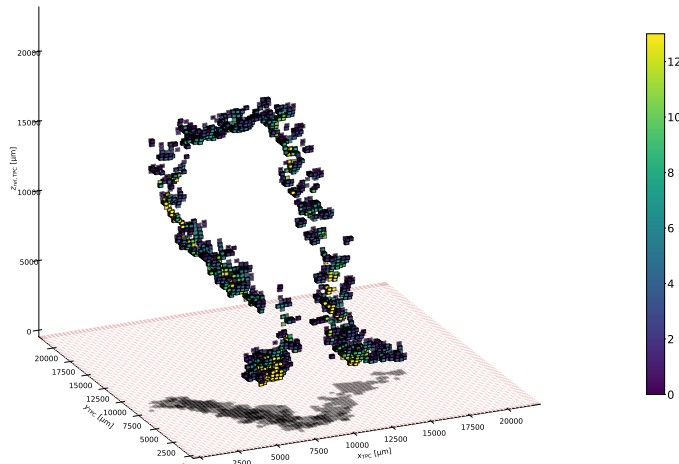

Figure 1: A 40 keV electron recoil reconstructed in 3D using the gas TPC in Jaegle et al. (2019). Electron recoils can have complex shapes and ambiguous starting points. An agent must identify the start of the recoil and determine how much of it to fit to determine the initial direction accurately. The color scale depicts a measure of ionization referred to as the time over threshold (TOT).

We develop a deep learning approach to analyze 3D events and probabilistically predict their initial direction. To our knowledge, this is the first deep probabilistic approach for predicting 3D directions with the Von Mises-Fisher distribution on $S^2$. Our framework significantly outperforms traditional algorithms when applied to a simulated electron recoil data set. Leveraging our framework's uncertainty predictions, performance can be enhanced by discarding high-uncertainty samples. By discarding only the top percentile, our framework achieves the best expected directional performance (as defined in Appendix A). This development is far-reaching as it enhances the sensitivity of all directional recoil detection experiments. Exciting possibilities include aiding in the discovery of dark matter (Billard et al., 2014; O'Hare, 2021) or resolving the solar abundance problem (Villante & Serenelli, 2019; O'Hare et al., 2022). The framework can also be applied to general problems where direction needs to be predicted, we demonstrate this by applying it to the task of detecting 3D arrows directions in Appendix B.

Our contributions include: 1) The first deep learning framework that predicts direction probabilistically with the von Mises-Fisher distribution on $S^2$, 2) the associated loss function along with approximations that stabilize training, 3) two tests that empirically demonstrate calibration, something which is lacking in related work on predicting orientation, 4) a model capable of analyzing highly-segmented 3D data using state-of-the-art sparse convolution techniques.

## 2 RELATED WORK

### 2.1 RELATED WORK FROM THE PHYSICS COMMUNITY

Others have been interested in applying deep learning to directional detectors (Schueler et al., 2022; Glaser et al., 2023). The most related application is X-ray polarimetry Weisskopf et al. (2016) where it is necessary to determine the initial direction of an electron track. A key difference is that they reconstruct tracks in 2D and hence only need to predict directions on $S^1$. In the work of Kitaguchi et al. (2019), the task is framed as a classification problem. The unit circle is split into 36 segments and a convolutional neural network is trained to classify tracks into a segment via the cross entropy loss. Event selection is performed based on the predicted probabilities. Our approach is capable of predicting directions continuously and is also able to do event selection, using the predicted

uncertainties. The work of Peirson et al. (2021) uses a deep ensemble to determine 2D directions with estimates for both data and model uncertainty. The model uncertainty is estimated by using an ensemble of models. The data uncertainty is modeled by minimizing the negative log-likelihood (NLL) of a probabilistic prediction. However, they use a Gaussian NLL instead of a von Mises NLL which is more suitable for 2D directions. Our NLL is derived from the von Mises-Fisher distribution on $S^2$, the appropriate probabilistic model for learning 3D directions.

## 2.2 RELATED WORK FROM THE MACHINE LEARNING COMMUNITY

In computer vision, several approaches have applied deep networks to estimating 3D orientation from 2D images (Liao et al., 2019; Huang et al., 2018; Xiang et al., 2018; Mahendran et al., 2017; 2018). Only a handful consider modeling orientation uncertainty (Prokudin et al., 2018; Gilitschenski et al., 2020; Mohlin et al., 2020). All of these approaches are concentrated on predicting orientation. This is similar to predicting direction but different. Orientation includes an additional dimension, which would introduce a spurious degree of freedom for our task. For example, an airplane's direction is naturally represented by a point on $S^2$, while the airplane's orientation requires an additional parameter specifying a rotation about the plane's main axis. Representing probability distributions over orientations in 3D is non-trivial, and the approaches of Prokudin et al. (2018); Gilitschenski et al. (2020); Mohlin et al. (2020) are all different.

Prokudin et al. (2018) uses Euler angles (pan, tilt, roll) to represent orientation. A unidimensional von Mises distribution over each angle captures orientation uncertainty. However, jointly predicting Euler angles has problems due to degeneracies known as gimbal lock. In Mohlin et al. (2020) orientation is represented by 3D rotation matrices. The orientation uncertainty is estimated by outputting a distribution over $SO(3)$, parameterized by the matrix Fisher distribution. However, this method poorly models classes with rotational symmetries, such as the test case in Appendix B. Gilitschenski et al. (2020) uses unit quaternions as a representation for object orientation. The uncertainty is modeled using the Bingham distribution, an antipodally symmetric distribution over $S^3$. Unit quaternions do not suffer from gimbal lock and are more compact than rotation matrices. However, the computationally expensive Bingham normalization constant necessitates backpropagation through an interpolator and use of a lookup table.

Our task is not to find the orientation of a rigid body. We want to find the initial direction in the trajectories imaged by our detectors. The direction lives on $S^2$, so frameworks that output distributions over $SO(3)$ or $S^3$ are inappropriate and fail to leverage a dimension of symmetry. This work proposes an approach focused on predicting directions with distributions on $S^2$. Predicting direction is a general task with many use cases beyond directional recoil detection, including predicting where something is pointing, predicting where someone is looking, determining the source direction of a signal, and predicting wind directions from video images.

## 3 THE MODEL

We develop two deep learning models. The first is the heteroscedastic convolutional neural network (HCN), a probabilistic model that predicts directional distributions on $S^2$. The second is the regular convolutional neural network (RCN), a deterministic model that only estimates direction. The RCN model serves as deep learning baseline which the HCN model can be compared to.

## 3.1 ARCHITECTURE

The input is a (120,120,120,1) voxel grid, which means every event has 1,728,000 features. For a typical electron recoil simulation (discussed in section 4) the fraction of non-zero features is on the order of $10^{-4}$. Sparsity is a common feature in highly-segmented 3D data and it is essential to take advantage of it to keep computational requirements feasible. All data is stored as `PyTorch Sparse Tensors` in the COO(rdinate) format, where only the non-zero entries and their indices are stored. A DataLoader is used to load batches of sparse tensors on the fly. The coordinates and values of the non-zero features are passed into our models, where they are immediately converted into a `spconv.SparseConvTensor` (Spconv Contributors, 2022).

Both models begin with a feature extraction portion, a series of submanifold sparse convolution, sparse convolution, and sparse max pooling layers. The details of the layers and the order in which they are applied is outlined in Figure 2. These sparse operations allow us to convolve and downsample our input data without expressing them as dense tensors, significantly increasing speed and reducing memory usage. Sparse convolution and max pooling are equivalent to their dense counterparts, except they operate on sparse tensors. In submanifold sparse convolution (Graham & van der Maaten, 2017; Graham et al., 2018), padding is applied so that the input and output have the same shape. An output site is active if and only if the corresponding input site is active, in which case the output feature vector is computed in the same manner as for regular convolution. Since the number of active sites is unchanged, submanifold sparse convolution allows us to process our input through several layers while maintaining its sparseness. The implementation of the layers specified in Figure 2 is through `SpConv` (Spconv Contributors, 2022) which is based on Yan et al. (2018).

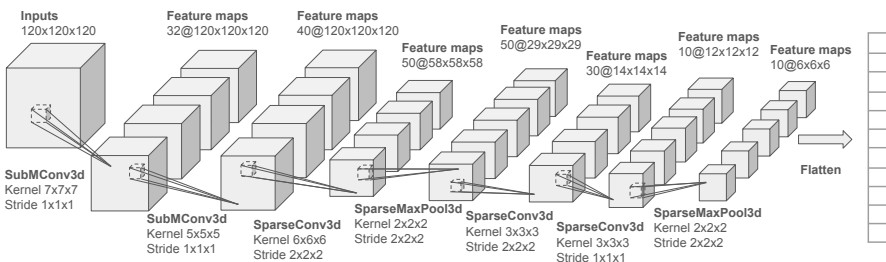

Figure 2: The feature extraction portion of our models. Each convolutional layer includes a learnable bias and is followed by a RelU activation. Padding is only applied in the SubMConv3d layers, where it is set such that the input and output have the same shape. The implementation is via `SpConv` (Spconv Contributors, 2022).

After applying the layers in Figure 2, the data is converted into a dense tensor and flattened. The flattened tensor is then passed on to the dense portion of the neural network. The HCN and RCN models have the same feature extraction portion but differing dense portions. For the RCN model, the dense portion is illustrated on the left of Figure 3. Here, the dense portion is fully connected and the output is 3 normalized neurons, to be interpreted as the predicted direction on $S^2$. The dense portion of the HCN model is illustrated on the right of Figure 3. The HCN model is a copy of the RCN model with an additional head. The additional head leads to a single output neuron to be interpreted as the predicted uncertainty. The details of the dense portions of our models are in Figure 3.

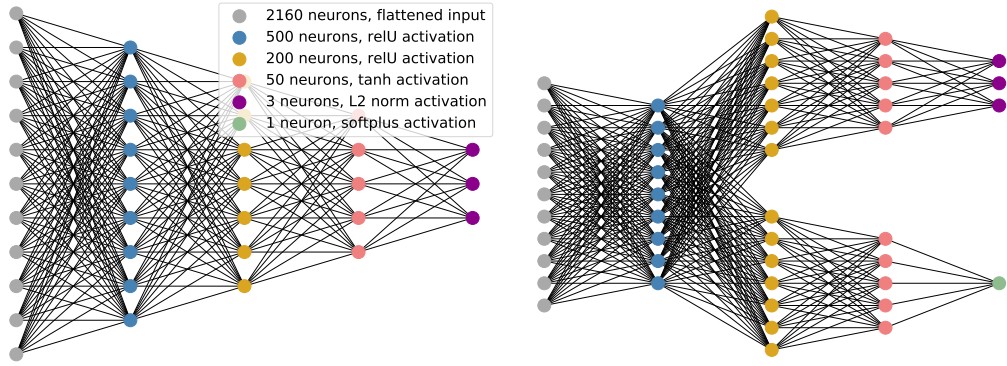

Figure 3: The dense portion of our models. The feature extraction described in Figure 2 is followed by either the left or the right neural network to make the RCN or HCN model, respectively. The illustrations do not display the actual number of neurons per layer. The number of neurons and the activation functions are specified in the legend. A learnable bias is added to every layer.

## 3.2 Loss functions

The RCN model is trained using the cosine distance loss function

$$\text{CSLoss} = \frac{1}{N} \sum_{i=1}^{N} 1 - \frac{\boldsymbol{y}_i \cdot \boldsymbol{y}_{pred_i}}{\max(|\boldsymbol{y}_i||\boldsymbol{y}_{pred_i}|, \epsilon)}, \tag{1}$$

where $\boldsymbol{y}_{pred_i}$ is the predicted direction, $\boldsymbol{y}_i$ is the true direction, $\epsilon$ is a small positive value to avoid division by $0$, and $N$ is the number of examples in a batch.

The HCN model is trained via heteroscedastic regression to predict a distribution of directions by specifying a mean direction and spread (uncertainty). To do this we need a probabilistic distribution of directions on $S^2$. The 5-parameter Fisher–Bingham distribution (Kent, 1982) is the analogue of the bivariate normal distribution on $S^2$

$$f(\boldsymbol{x}) = \frac{1}{c(\kappa, \beta)} \exp\left\{\kappa \boldsymbol{v}_1^T \cdot \boldsymbol{x} + \beta\left[(\boldsymbol{v}_2^T \cdot \boldsymbol{x})^2 - (\boldsymbol{v}_3^T \cdot \boldsymbol{x})^2\right]\right\}. \tag{2}$$

Above, $c(\kappa, \beta)$ is a normalization constant, $\kappa \in \mathbb{R}_+$ determines the spread, $\beta \in [0, \kappa/2)$ determines the ellipticity, and $\boldsymbol{v}_1, \boldsymbol{v}_2, \boldsymbol{v}_3$ determine the mean direction, major axis, and minor axis, respectively. We simplify Equation 2 to only model uncertainty isotropically about the mean direction by setting $\beta = 0$. By doing so, we obtain

$$f(\boldsymbol{x}) = \frac{\kappa}{4\pi \sinh \kappa} \exp\left(\kappa \boldsymbol{v}_1^T \cdot \boldsymbol{x}\right). \tag{3}$$

This special case of the 5-parameter Fisher–Bingham distribution is also known as the von Mises–Fisher distribution on $S^2$ (Mardia et al., 2000).

Given the predicted directions ($\boldsymbol{y}_{pred_i}$) and predicted uncertainties ($\kappa_i$) of the HCN model, as well as the true directions ($\boldsymbol{y}_i$), we use Equation 3 to calculate the likelihood as

$$\mathcal{L} = \prod_{i=1}^{N} \frac{\kappa_i}{4\pi \sinh(\kappa_i)} e^{\kappa_i(\boldsymbol{y}_i \cdot \boldsymbol{y}_{pred_i})},$$

and the NLL, scaled by batch size, is

$$\frac{-\ln \mathcal{L}}{N} = \frac{-1}{N} \sum_{i=1}^{N} \ln\left(\frac{\kappa_i}{4\pi \sinh(\kappa_i)}\right) + \kappa_i(\boldsymbol{y}_i \cdot \boldsymbol{y}_{pred_i}). \tag{4}$$

The HCN model is trained to predict directional distributions as in Equation 3 that minimize the NLL.

We note that Equation 4 generalizes Equation 1. If we assume $\kappa_i$ is constant (the uncertainty is uniform), the first term in Equation 4 becomes a constant. Furthermore, Since $\boldsymbol{y}_{pred_i}$ and $\boldsymbol{y}_i$ are unit vectors, the denominator in Equation 1 is $1$. Now, it is easy to see that under the uniform uncertainty assumption, Equations 1 and 4 are equivalent up to a constant offset and scale factor. Therefore, the HCN and RCN models are heteroscedastic/homoscedastic counterparts.

## 3.3 Practical considerations

Now that we've outlined the models, we discuss the practical considerations of implementing the HCN model. The first is the ln term on the right-hand side of Equation 4. If $\kappa_i \geq 87$, the argument of the ln term evaluates as zero making the loss infinite. When $\kappa = 0$, Equation 3 is isotropic, and as $\kappa$ increases the distribution becomes more localized (the uncertainty decreases). Hence, as the model learns directions accurately it will encounter $\kappa_i \geq 87$, and all gradients will be `nan`. To avoid this, we use the limit

$$\lim_{\kappa_i \to \infty} ln\left(\frac{\kappa_i}{4\pi \sinh(\kappa_i)}\right) = ln\left(\frac{\kappa_i}{2\pi}\right) - \kappa_i.$$

This limit is a close approximation, even for small values of $\kappa_i$. If $\kappa = 9$, the values of $ln(\frac{\kappa_i}{4\pi \sinh(\kappa_i)})$ and $ln(\frac{\kappa_i}{2\pi}) - \kappa_i$ are indistinguishable when using the 32-bit floating point data type. Therefore, instead of using Equation 4, we express our loss function as

$$NLL = \frac{-1}{N} \sum_{i=1}^{N} \text{where} \left(\kappa < 9, \ln\left(\frac{\kappa_i}{4\pi \sinh(\kappa_i)}\right), ln\left(\frac{\kappa_i}{2\pi}\right) - \kappa_i\right) + \kappa_i(\boldsymbol{y}_i \cdot \boldsymbol{y}_{pred_i}). \tag{5}$$

Above, we use the PyTorch operation `torch.where` which has three arguments. If the condition in the first argument is True then `torch.where` is evaluated as the second argument; otherwise, `torch.where` is evaluated as the third argument.

Using `torch.where` introduces another consideration. If the second or third argument is infinite, then `torch.where` will produce nan in the backward pass, even if the condition selects the non-infinite term. This is discussed in `https://github.com/pytorch/pytorch/issues/68425`. The PyTorch MaskedTensor is suggested as a solution for this (see `https://pytorch.org/maskedtensor/main/notebooks/nan_grad.html`); however, this API is still in the prototype stage and is not implemented for all of the functions we use. Therefore, Equation 5 does not solve our issue because we still get nan in the backward pass whenever $\ln(\frac{\kappa_i}{4\pi\sinh{(\kappa_i)}})$ evaluates as infinity.

To solve our problem, we must remove the term $\ln\frac{\kappa_i}{4\pi\sinh{(\kappa_i)}}$ from Equation 5 completely. We replace it with its 15th-order Taylor Series about $\kappa = 0$

$$T_{15}(\ln\frac{\kappa_i}{4\pi\sinh{(\kappa_i)}}) = \ln(4\pi) + \frac{\kappa^2}{6} - \frac{\kappa^4}{180} + \frac{\kappa^6}{2835} - \frac{\kappa^8}{37800} + \frac{\kappa^{10}}{467775} - \frac{691\kappa^{12}}{3831077250} + \frac{2\kappa^{14}}{127702575}.$$

We stop at the 15th-order because higher powers of $\kappa$ make the Taylor series unstable. We now have two approximations for $\ln\frac{\kappa_i}{4\pi\sinh{(\kappa_i)}}$, for low and high $\kappa$. Putting these together we arrive at the final form of the loss function for the HCN model

$$NLL = \frac{-1}{N}\sum_{i=1}^{N}\text{where}\left(\kappa < 2.65, T_{15}(\ln\frac{\kappa_i}{4\pi\sinh{(\kappa_i)}}), ln(\frac{\kappa_i}{2\pi}) - \kappa_i\right) + \kappa_i(\boldsymbol{y}_i \cdot \boldsymbol{y}_{pred_i}). \quad (6)$$

The condition $\kappa < 2.65$ is chosen so that the maximum percent difference between Equations 4 and 6 is minimized to $0.14\%$.

## 4 APPLICATION TO ELECTRON RECOILS

For a demonstration of our models on a simple case see Appendix B. Here, we apply our models to determine the initial direction of electron recoils because that is the most complex scenario in gas TPCs. As an electron recoils through gas it undergoes multiple scattering which alters the trajectory of the recoiling electron and creates a track of secondary, ionized electrons. By reconstructing the positions of the secondary electrons in 3D, directional gas TPCs can infer the initial direction of the recoiling electron. Determining the direction is a complicated task because of the non-trivial track shapes, illustrated in Figure 1. An agent must determine which side of the recoil track is the starting point, and how much of the track beyond the starting point to fit in order to optimize the initial direction prediction. As the electron recoils it loses energy and in the process becomes more highly ionizing. Hence the charge density along the track can be used to determine the starting point. To first order, there are two effects influencing the angular resolution of electron recoils: the multiple scattering of the recoiling electron and the effective point resolution with which the secondary electrons are detected. Multiple scattering makes the electron lose directionality as it travels, suggesting only the very beginning of the track should be used to determine the initial direction. On the other hand, the effective point resolution sets a lower limit on the length scale where meaningful information can be extracted from the track. The models need to learn the best trade-off between these two effects on a tack-by-track basis to provide the most accurate initial direction predictions.

### 4.1 SIMULATION, PREPROCESSING, AND DATA SPLITS

Below we detail the steps taken to create our simulated electron recoil data sets. The specification of the gas mixture, pressure, temperature, and diffusion are inspired by Jaegle et al. (2019).

1. We simulate $10^6$ electron recoils at 40, 45, and 50 keV in a $70\%$ He : $30\%$ $CO_2$ gas mixture at $20°C$ and 760 Torr using Degrad (Biagi, 2014). All electron recoil simulations begin at the origin with the initial momentum in the positive z-direction. This produces a track of secondary (ionized) electrons for each electron recoil simulation, illustrated in Figure 4a.

2. To include the diffusion in the detector, Gaussian smearing is applied to each ionized electron in the track. For each track, the amount of smearing is drawn from a uniform distribution of $160 - 466$ $\mu$m, the smallest and largest expected values in Jaegle et al. (2019).

3. To make our simulations isotropic, a random rotation is applied to the track. The true initial direction after rotation is saved.

4. The track is translated so that the origin is the center of charge. An example of a simulation at this stage is shown in Figure 4b

5. The tracks are then binned into a $(120, 120, 120)$ voxel grid, each voxel is a $(500\mu m)^3$ cube. In binning, data is directly transformed into `PyTorch Sparse Tensors` in the COO format. Tracks that are not fully contained in the voxel grid are discarded. An illustration of a simulation at this stage is shown in Figure 4c.

The final dataset contains 2,766,798 simulations. A random data split is used to create a training set (80%) and a validation set (20%). To create the test data sets we simulate another $2 \times 10^4$ electron recoils at 40 and 50 keV. We follow the same steps outlined above except for a slight modification. The applied Gaussian smearing is no longer drawn from a uniform distribution. Instead, we probe two specific cases: a high diffusion case where $443$ $\mu$m of smearing is applied and a low diffusion case where $200$ $\mu$m is applied. The two energies and two diffusion cases give us four test data sets.

The simulation framework, `Degrad`, is the standard software package used in physics for simulating the interaction of low-energy electrons in gasses Pfeiffer et al. (2019). The focus of this work is to compare the different models, and our simulations provide a typical setup on which to compare performance. The performance in any particular detector is expected to differ slightly due to the sim-to-real gap. In the future we plan to use radioactive source data to train our models, thereby avoiding the sim-to-real gap entirely. Simulation parameters, such as diffusion and binning, were deliberately conservative compared to what is experimentally achievable. We verified that model performance is robust against modest variation of the Degrad input parameters, such as temperature.

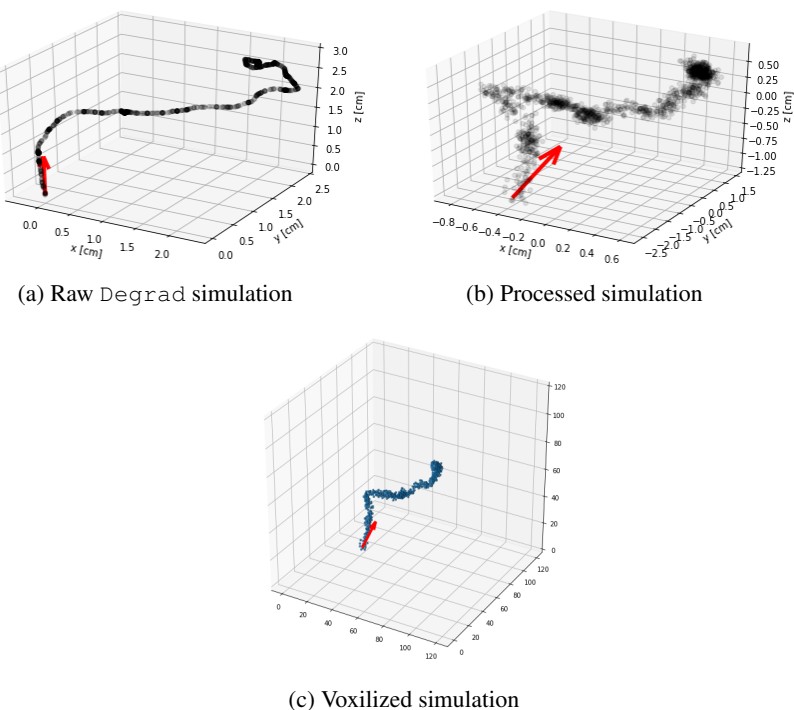

(a) Raw `Degrad` simulation          (b) Processed simulation

(c) Voxilized simulation

Figure 4: An electron recoil simulation at various prepossessing stages. The processed simulation is made by randomly rotating, applying Gaussian smearing, and mean-centering the raw simulation. That is then binned into a $(120, 120, 120)$ voxel grid with a $(500\mu m)^3$ resolution. The red arrows indicate the true direction.

## 4.2 TRAINING

Both models are trained using the `Adam` optimizer with lr $= 0.0001$, betas $= (0.94, 0.999)$, eps $= 10^{-7}$, and the `PyTorch` default value for all other parameters. The deep learning models are trained with mini-batches of size $N = 256$. Hyperparameters were optimized by exploring values in the range eps $\in [10^{-8}, 10^{-6}]$, lr $\in [10^{-6}, 10^{-4}]$, and $N \in [64, 256]$.

Both the RCN and HCN models are trained on the training set while their loss on the validation set is used for early stopping and hyperparameter selection. If the validation loss has not decreased in the last five epochs, training is stopped. The weights corresponding to the lowest validation loss are saved and used for the final model. The validation cosine distance loss achieved by the final RCN model is 0.078 and the validation NLL loss achieved by the final HCN model is 0.179.

## 4.3 PERFORMANCE

We evaluate the trained model on the independent test sets discussed in Section 4.1 by comparing it to a non-machine learning algorithm and to our estimation of the best expected directional performance. The non-machine learning algorithm (Non-ML) is adapted from Marco et al. (2022). We generalize the original algorithm to 3D and explore different variations where we tune its parameters on the test sets and give the algorithm the correct head-tail. The best expected directional performance (Best-Expected) is estimated by utilizing information that is only available in simulation. The purpose of Best-Expected is not to be a viable option for analysis but to provide a benchmark of exceptional performance. Non-ML and Best-Expected are detailed in Appendix A. Furthermore, the RCN model provides a deep learning baseline to compare HCN to.

To compare all of our models, we use the cosine distance loss in Equation 1. In Appendix A we explain that the Non-ML algorithm has a track efficiency based on the parameters used. Hence, when comparing the models we plot the cosine distance loss versus the percentage of omitted tracks, referred to as the efficiency cut. In Figure 5, we display the performance of all models on the 40 keV test data set with 443 $\mu$m Gaussian smearing (left) and the 50 keV test data set with 200 $\mu$m Gaussian smearing (right). In both cases, we find that the deep learning models significantly outperform Non-ML, even under the most optimistic assumptions. On the 40 keV test data set with 443 $\mu$m Gaussian smearing and at a $0\%$ efficiency cut, the RCN, HCN, and Best-Expected models achieve a cosine distance loss of 0.104, 0.104, and 0.098, respectively. On the 50 keV test data set with 200 $\mu$m Gaussian smearing and at a $0\%$ efficiency cut, the models achieve a cosine distance loss of 0.0624, 0.0632, and 0.0569, in the same order. The HCN and RCN models have similar performance. However, the accuracy of the HCN model can be improved by discarding events with low $\kappa_i$ (high predicted uncertainty). The HCN model outperforms Best-Expected with just a $1\%$ efficiency cut, in both cases.

To assess the calibration of the HCN model we plot a 2D histogram of the predicted uncertainty ($\kappa_i$) and the angle from the true direction to the predicted direction ($\theta_i$) for all test data, displayed in Figure 6a. The spread in $\theta_i$ gets smaller for higher values of $\kappa$, indicating that the HCN model is predicting $\kappa_i$ appropriately. The solid red curve in Figure 6a shows the mean angle, calculated for each bin in $\kappa$. Using Equation 3, we can also calculate the expected mean angle as a function of $\kappa$

$$\theta_{\mathrm{avg.}} = \int_0^{2\pi} \int_0^{\pi} \theta \frac{\kappa}{4\pi \sinh \kappa} \exp\left(\kappa \cos\left(\theta\right)\right) \sin(\theta) d\theta d\phi = \frac{\pi}{2\sinh(\kappa)} \left(I_o(\kappa) - e^{-\kappa}\right),$$

where $I_o$ is a modified Bessel function of the first kind. The expected mean angle as a function of $\kappa$ is plotted as the dashed black line in Figure 6a. The agreement of the expected mean angle with the mean angle for each bin in kappa suggests that the HCN model is well-calibrated.

Building onto the 2D histogram in Figure 6a, we check whether the predicted directions in a $\kappa$ bin behave as if they are drawn from a von Mises-Fisher distribution with that value of $\kappa$. For each bin in $\kappa$, we compute a distribution of $\boldsymbol{y}_i \cdot \boldsymbol{y}_{pred_i}$ values and fit it with Equation 3 to obtain $\kappa_{\mathrm{fit}}$. In a perfect scenario, we expect $\kappa_{\mathrm{fit}}$ to match the bin center value. The $\kappa_{\mathrm{fit}}$ value for each bin versus the bin center value in plotted in Figure 6b. The diagonal solid line indicates perfect agreement and the dashed lines around it show the width of a bin in $\kappa$. Looking at both Figures 6a and 6b, we see good agreement in the predicted and fit $\kappa$ values in the regions where the majority of our statistics lie.

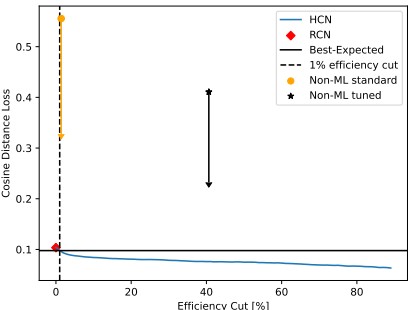 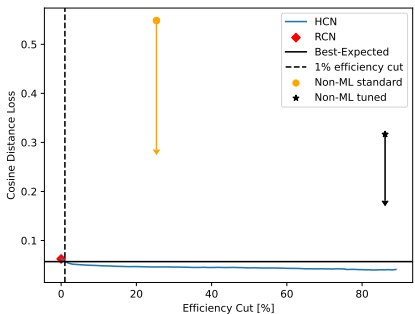

Figure 5: Cosine distance loss versus efficiency cuts for all models on the $40$ keV test data set with $443\,\mu$m Gaussian smearing (left) and the $50$ keV test data set with $200\,\mu$m Gaussian smearing (right). The RCN (red diamond) and Best-Expected (horizontal line) models have no efficiency cuts. For the HCN model, we improve accuracy by cutting examples with higher predicted uncertainty, results are presented with a blue curve. The dashed vertical line indicates when the HCN model beats Best-Expected. The orange point and black star indicate the performance of the Non-ML method with standard parameters and with parameters tuned on the test set, respectively. The downward arrows indicate how much improvement is possible if we cheat and give Non-ML the correct head-tail.

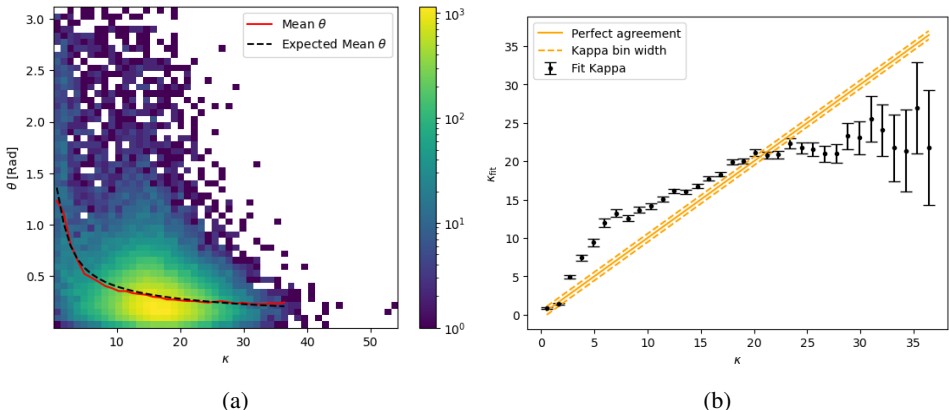

(a)                                          (b)

Figure 6: Left: 2D histogram of predicted uncertainty ($\kappa$) and the angle from true to predicted direction for all test data. The red curve is the mean angle for each bin in $\kappa$. The black dashed curve is the expected mean angle according to Equation 3. Right: The cosine of the angles for each bin in $\kappa$ is fit to Equation 3 to obtain $\kappa_{\text{fit}}$, plotted with statistical error bars versus the bin center $\kappa$ value.

## 5    CONCLUSION

This works integrates all the essential components required for applying deep learning to modern directional recoil detectors. We discuss the challenge of handling sparse 3D data, which is increasingly critical as detectors become more finely segmented. For the first time, we introduce a deep probabilistic approach that utilizes the von Mises-Fisher distribution on $S^2$ to predict 3D directions along with their associated uncertainties. We outline the approximations on the NLL necessary for stable training. Additionally, we propose two tests to assess model calibration and we demonstrate that efficiency cuts on high uncertainty events effectively improve accuracy. The culmination of our work is a general and robust framework that marks a significant advancement over existing non-machine learning methods. These advancements have the potential to contribute to several realms of fundamental physics. This general approach is applicable to any task requiring 3D direction estimation with uncertainty.

ACKNOWLEDGMENTS

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

## A    NON-MACHINE LEARNING ALGORITHMS

Here, we detail the non-machine learning algorithms to which we compare our models. The first algorithm, Non-ML, is adapted from Marco et al. (2022) where it is discussed in the context of determining the initial direction of a photoelectron track imaged in 2D by the IXPE. We generalized the algorithm so that it can be applied to our 3D simulations. The Non-ML algorithm is outlined below:

1. Find the barycenter ($\boldsymbol{x}_b$) of the track, defined as $\boldsymbol{x}_b = \frac{\sum_i q_i \boldsymbol{x}_i}{\sum_i q_i}$ , where $q_i$ is the charge in a voxel and $\boldsymbol{x}_i$ is the position of the voxel. Center the track on $\boldsymbol{x}_b$.

2. Compute the weighted covariance matrix. Use its singular value decomposition (SVD) to find the principal axis $\boldsymbol{v}_{\mathrm{PA}}$ (the axis on which the second moment is maximized) and the second moment along $\boldsymbol{v}_{\mathrm{PA}}$, denoted as $M_2$.

3. Compute the third moment ($M_3$) about $\boldsymbol{v}_{\mathrm{PA}}$. $M_3$ is also known as the skewness and it is used to keep only the initial part of the recoil track. Keep only the points satisfying $sgn((\boldsymbol{x}_i - \boldsymbol{x}_b) \cdot \boldsymbol{v}_{\mathrm{PA}}) = sgn(M_3)$.

4. $M_2$ sets the length scale along the principal axis, keep only the points satisfying $1.5M_2 < (\boldsymbol{x}_i - \boldsymbol{x}_b) \cdot \boldsymbol{v}_{\mathrm{PA}} < 3M_2$

5. The two conditions above are meant to isolate the beginning portion of the track. The interaction point ($\boldsymbol{x}_{\mathrm{IP}}$) is defined as the charge-weighted center of the remaining points. Center the remaining points on $\boldsymbol{x}_{\mathrm{IP}}$.

6. The remaining points are re-weighted with $w_i = \exp\left(-dist(\boldsymbol{x}_{\mathrm{IP}}, \boldsymbol{x}_i)/w_o\right)$, where $w_o = 0.05$ as specified in Marco et al. (2022). Compute the weighted covariance matrix and use SVD to find the principal axis, this gives the initial axial direction of the track, denoted as $\boldsymbol{v}_{\mathrm{IP}}$.

Above, $1.5M_2$, $3M_2$, and $w_o = 0.05$ are all adopted from Marco et al. (2022), it is possible that these values are not optimal for our application. Hence, we also investigate the performance when these parameters are tuned to minimize the cosine distance loss on each test data set. There is no discussion of assigning a head-tail to $\boldsymbol{v}_{\mathrm{IP}}$. We employ two approaches: In the first, we assign the head-tail such that $\boldsymbol{v}_{\mathrm{IP}} \cdot \boldsymbol{v}_{\mathrm{PA}} => 0$. In the Second, we simply assign the correct head-tail, such that $\boldsymbol{v}_{\mathrm{IP}} \cdot \boldsymbol{v}_{\mathrm{True}} => 0$ where $\boldsymbol{v}_{\mathrm{True}}$ is the true direction. Finally, the selections applied in steps 3 and 4 may not leave enough points to determine a principal axis for a subset of the tracks. In this case, the track is omitted and the track efficiency is noted. When tuning the parameters, we only consider cases resulting in a track efficiency of greater than $10\%$.

The second algorithm, Best-Expected, attempts to estimate the best possible directional performance by using information that is only available in simulation. The purpose is not to be a viable option for analysis but to provide a benchmark allowing us to check how close our models are to the best expected performance. The implementation of this method is outlined below:

1. For each simulation, Identify the true starting point. The true starting point is information that is only known in simulation. One of the key challenges faced by our models is identifying the starting point which can be ambiguous.

2. Make a sphere of radius $\varepsilon$ centered on the true starting point. The value of $\varepsilon$ is specified by tuning it to minimize the cosine distance loss for each test data set.

3. Determine the principal axis of the points contained within the sphere using SVD.

4. Assign a head-tail to the principal axis such that it agrees with the true direction. In this step, the true label is used to assist the algorithm, giving it a significant advantage.

5. The final vector direction obtained is used as the predicted direction of this method.

Since this algorithm is given information about the true starting point, it is tuned on the test data sets, and it uses the truth head-tail information, we reason it is a good approximation of the best expected performance.

## B  APPLICATION TO A SIMPLE CASE

To provide an illustrative tutorial of applying our models, we demonstrate their application to determining the direction of 3D arrows. Isotropic arrows with a constant shape and uniform density are generated and binned into a $(120, 120, 120)$ voxel grid. An example is displayed in figure 7. We generate 6000 arrows, 4000 are used for the training set and 1000 are used for the validation and test sets. The RCN and HCN models are trained on the training set while their validation loss is used for early stopping. If the validation loss has not decreased in the last 2 epochs, training is stopped. The weights corresponding to the lowest validation loss are saved and used for the final model.

In this simple case, both models are fully trained within minutes. The cosine distance loss of the RCN and HCN models on the test data set is $4 \times 10^{-4}$ and $8 \times 10^{-4}$, respectively. As expected, the direction predictions are much more accurate than Section 4. The lowest predicted $\kappa_i$ by HCN on the test data set is $443$, meaning the model is much more confident in the predicted directions than for the electron recoil case. The code needed to create the arrow data sets, train our models, and test our models is available at `https://github.com/fake-acc/tmp-fake-repo`.

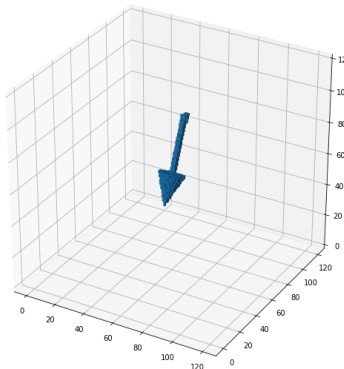

Figure 7: An example of a 3D arrow for the application to a simple case.

