# OpenReview forum: "Deep probabilistic 3D angular regression for directional dark matter detectors"
_ICLR.cc/2024/Conference — Submitted to ICLR 2024_

### Official Review · Reviewer_qwYh · 2023-10-22

**Soundness:** 3 good
**Presentation:** 1 poor
**Contribution:** 2 fair
**Rating:** 3
**Confidence:** 3

**Summary:**

This paper presents a probabilistic deep learning model designed to predict the 3D directions of elementary particles from point cloud data, specifically targeting the detection of dark matter. The primary challenge tackled is determining the initial directions of low-energy electrons. The architecture used is a sparse 3D convolutional neural network. The output is parameterized using a von Mises-Fisher distribution, as the directional outputs need to be constrained to the 2-sphere. When compared against a non-machine learning approach on a simulated dataset, the proposed model shows a significant improvement in mean cosine distance.

**Strengths:**

S1. Method is appropriate: The idea of employing a probabilistic deep learning model for analysing particle trajectories has promise. I also like the fact that the proposed approach uses both a probabilistic output for quantifying the uncertainty as well as doing on-manifold predictions.

S2. Model Calibration: The paper shows that the model is well-calibrated on simulated data and can identify low-uncertainty samples, which is crucial for real-world applications. This ability to filter out ambiguous predictions is very useful for improving the accuracy and realiability which is important in science applications.
The comparison to a deterministic model trained with the cosine distance loss also demonstrates that the method performs well compared to the baseline.

S3. Technical Depth: The method seems to be well thought out and goes into detail on computing the negative log-likelihood loss, and practical considerations for stable training.

**Weaknesses:**

W1. Limited Comparison: While the paper compares the proposed method with a non-machine learning algorithm, it lacks a comprehensive comparison against other deep learning-based methods, especially those for predicting uncertainties on manifolds. See, for example [1] and the references therein.

W2. Real-world Evaluation: The paper primarily relies on a simulated test set. Although simulations are valuable, as this is an application-focussed paper, it would be useful to also evaluate the model on real-world data, possibly obtained from actual detectors, to gauge its performance under realistic conditions.

W4. Interest to the community: The paper is focussed on a very specific application and I'm not sure how relevent the method is to the ICLR community. Perhaps it would be better suited to a physics journal. Related to this, if the paper is aimed at the ICLR audience, then the introduction needs to be written in a way that clearly explains the problem and why it is important from an ML perspective.

[1] Gilitschenski, Igor, et al. "Deep orientation uncertainty learning based on a bingham loss." International conference on learning representations. 2019.

**Questions:**

Recommendations:

1. I would like the authors to explain why this method is relevant to the machine learning community and also give a better overview of why the problem is important (so that non-particle physiscists can understand its significance).

2. The authors should also consider comparing their method with other machine learning or deep learning methods, if available, that also target on-manifold uncertainty predictions. This will help position the contribution better with respect to existing work.

3. Further analysis of the model's generalizability could be insightful. This will help readers gauge the model's applicability to other problems within the broader particle detection domain.

---

> ### Author Response · Authors · 2023-11-20
>
> Thank you very much for the insightful feedback. Below, we address the stated weaknesses and recommendations.
>
> # In response to W1 and recommendation 2
>
> We have clarified the relationship to prior work. Please see “Comparison to related work” in our general response.
>
> # In response to W2
>
> Please see the “Real-World Evaluation and the sim-to-real gap” discussion in our general comment.
>
> # In response to W4 and recommendation 1
>
> Please see the “Interest to the community / key Novelties” discussion in our general comment.
>
> # In response to recommendation 3
>
> This is a very general approach. In appendix B (previously Section 5), we demonstrate the model on a toy example with arrows in 3D. For the electron recoils, the shape of the ionization track is unique for each sample and highly variable. The shapes also depend heavily on the energy of the electron recoils and the amount of gaussian smearing applied (a parameter of the detector setup), and in our experiments we have varied both of these to show that our model generalizes well.

---

> ### Author Response · Authors · 2023-11-28
>
> We appreciate the helpful feedback enhancing our paper.  We've discussed relevant work, including Gilitschenski, Igor, et al., clarifying the rationale for comparing our model to our own baselines. We’ve clarified why real-world evaluation isn't our primary focus at this stage, we've demonstrated our model's robustness to simulation parameter variations. Our contribution, the first deep probabilistic direction prediction model, is relevant to the ICLR community because it addresses a general problem. With the discussion period extended until Dec. 1, we welcome additional feedback to address any remaining concerns. If you feel that your concerns have been addressed, then we encourage you to reassess your rating.

---

### Official Review · Reviewer_7Dku · 2023-10-24

**Soundness:** 3 good
**Presentation:** 3 good
**Contribution:** 3 good
**Rating:** 8
**Confidence:** 3

**Summary:**

The paper introduces a deep learning framework for the estimation of 3D angles from input trajectories without timestamps. The application domain is particle physics, where detectors aim to probe electrically neutral particles and perhaps even detect dark matter through the recoil trajectories of electrically charged particles. The determination of the starting point and angle is crucial for rejecting noisy detections.

**Strengths:**

This is an interesting paper that formulates a solution to a problem in particle physics with a deep learning framework. The paper is presented in a way that is digestible to a non-physicist, with a very nicely explained introduction and motivation for a topic I am not familiar with. Both the proposed heteroscedastic network design and the von Mises Fisher distribution are novelties that improve the results in the simulated test cases.

**Weaknesses:**

I have two concerns with the paper that I will classify as "weaknesses".

1. While the results shown on simulated test cases are encouraging and show that the approach works in principle, it is unclear how large the sim-to-real gap is. It is hard to determine how realistic the Degrad simulations are as compared to real measurements of electron recoils. Furthermore, Degrad presumably carries with it a whole host of parameters; a few are mentioned in the text (e.g., gas mixtures, temperatures, pressure, etc.). While I understand that the combination of parameters chosen was inspired by prior work (Jaegle et al.) it is unclear how well these parameters are constrained in reality and what happens to the deep-learning model if it encounters variations such as those expected in real data. E.g., does the model completely break if we now test it on trajectories generated at 21° instead of 20°, etc.? These tests for systematic error sources won't determine the gap between simulation and reality, but they will be a first step towards that goal. I think that additional experiments explicitly testing these potential sources of systematic errors and/or a discussion that fully discloses these limitations should be present. The discussion should include what is key or missing from having this work on real (not simulated) data.

2. It is unclear from the paper exactly what the key novelty is. Initially, I thought that this is the first DL framework for the task, as is also demonstrated by the result comparison in Fig. 4. After reading the conclusion, it was unclear if there were prior DL works, and this paper introduces the von Mises Fisher distribution for prediction. It crucial to clearly clarify this in the text. If this is not the first DL approach to the task, I would expect a comparison with prior DL approaches.

Minor comments
--------------------
I have a few minor comments that also need addressing:

1. Section 5 probably should be moved to the supplementary, but in any case, it should not be after Section 4.3. Estimating an arrow might be a sanity check but it is irrelevant once the reader has already seen more complex examples.
2. What do the colors in Fig. 1 represent? Ionization? A color bar should be present and this should be mentioned in the caption
3. Consider replacing Table 1 with a more useful and digestible visual representation, like a block diagram
4. While the model includes an uncertainty estimate and is therefore heteroscedastic across different trajectories, it should be mentioned that the uncertainty is assumed to be homogeneous along a single trajectory (i.e. all data points within a trajectory have the same uncertainty)
5. The term "efficiency cut" is introduced in 4.3 (first paragraph) but only explained later in Fig. 4

**Questions:**

See weaknesses

---

> ### Author Response · Authors · 2023-11-20
>
> We sincerely appreciate this constructive feedback and address your concerns below:
>
> # In response to weakness 1
>
> Please see “Real-World Evaluation and the sim-to-real gap” in our general response.
>
> # In response to weakness 2
>
>  Please see “Interest to the community / key Novelties” and “Comparison to related work” in our general response for a discussion of our key novelties and how our work relates to similar tasks.
>
> # In response to the minor comments
>
> 1. We have moved Section 5 into the appendix.
>
> 2. Yes, the color scale represents ionization. A colorbar has been added and mentioned in the caption.
>
> 3. Table 1 has been replaced with a block diagram, Figure 2.
>
> 4. To clarify, our model predicts the initial direction for a single event, where an event consists of a particle traveling through the detector and causing ionization at many points in 3D space.
> The reviewer might also be referring to the fact that our model represents uncertainty isotropically. This is captured in the statement just above Equation 3, “We simplify Equation 2 to only model uncertainty isotropically about the mean direction by setting \beta = 0.”
>
> 5. Thank you for catching this. We updated the sentence to, “... we plot the cosine distance loss versus the percentage of omitted tracks, referred to as the track efficiency cut”

---

> ### Author Response · Authors · 2023-11-28
>
> Thank you for your initial feedback and response to our comment. To recap our response to your concerns:
>
> 1. We've shown that our conservative simulations yield a robust model against modest variations in simulation parameters.
> 2. We’ve clarified that our framework is the first for the task of probabilistically predicting 3D direction
> 3. We've included a comprehensive discussion positioning our work with respect to the extensively studied problem of predicting orientation.
>
> Should you have any lingering concerns or additional insights, we welcome your valuable feedback as our discussion period now extends until Dec. 1. If you believe your issues have been resolved, we encourage you to reconsider your rating.

---

### Official Review · Reviewer_c5Tr · 2023-11-20

**Soundness:** 3 good
**Presentation:** 2 fair
**Contribution:** 2 fair
**Rating:** 5
**Confidence:** 2

**Summary:**

This paper addresses the problem of predicting the initial direction of particles from recoil trajectories. The problem is potentially useful in directional detection of particles and could be used for dark matter detection. The paper models the initial directional distribution as a von Mises-Fisher distribution an uses a sparse-convolution network to fit the model parameters using. The effectiveness of the proposed network is demonstrated through experiments on simulation data.

**Strengths:**

- The paper presents backgrounds, motivations, and problem setups clearly and is understandable by one with limited knowledge in physics.
- The use of 3D sparse convolution networks and NLL appear appropriate for the problem.
- Experiments show clear improvements over non-learning algorithms on simulated data.

**Weaknesses:**

- Lack of real-world experiments.

  The effectiveness of the proposed learning-based method is only verified on data from simulation. It also does not explain why the specific choices of simulation parameters, as describes in section 4.1, are determined. For example, would performing Gaussian smearing on the simulation data favor the Gaussian-like von Mises-Fisher model, and it is not clear whether the proposed model is still a good choice for real-world data.

- Technical contribution to the learning community.

  The 3D sparse convolution and von Mises-Fisher probabilistic model are well-established techniques and do not seem to provide too much insight for the learning community. This work may be better suited for a conference/journal in physics.

  The work would be more valuable if it studies the effectiveness of different 3D learning architectures and probabilistic models when applied to real-world electron recoil data, or demonstrate the sim-to-real transferablility of such models.

**Questions:**

Minor questions comments:
- Some detailed information in section 3, such as 3.4, would better fit in the experiment section(e.g., section 4.2).
- What considerations are taken to determine the specific simulation parameters are used in section 4.1? Why is the test set different from the training set and how are the specific parameters determined?

---

> ### Author Response · Authors · 2023-11-22
>
> Thank you for you valuable time and feedback. We address the weaknesses and questions below.
>
> # Re: Lack of real-world experiments.
>
> Under “Real-World Evaluation and the sim-to-real gap” in our general response, we discuss the scope of our paper.
>
> Please also see “General comments about sim-to-real gap and systematic errors” where we elaborate on the sim-to-real gap.
>
> The specific choice of our simulation parameters is based on Jaegle et al. (2019). We have slightly modified the text on Section 4.1 to emphasize this further. The choice of binning segmentation was deliberately conservative with respect to what is experimentally achievable.
>
> Gaussian smearing, which models the diffusion, is not the only thing influencing the directionality of the track. The multiple scattering that is simulated by Degrad also influences directionality. Furthermore, diffusion is a Gaussian process, so if this process does favor the von Mises-Fisher distribution, then that motivates using the von Mises-Fisher distribution.
>
> Please also see our response “Re: systematic errors” where we demonstrate our model is robust against small fluctuations in the simulation parameters.
>
> # Re: Technical contribution to the learning community.
>
> While the von Mises-Fisher distribution is well-established, no other deep learning approaches utilize it in the way we do. We believe this work is well suited for a machine learning conference because
>
> 1. Predicting direction is a general problem.
> 2. To our knowledge, this is the first deep learning approach which probabilistically predicts 3D direction distributions
>
> Furthermore, our work includes other contributions such such as the approximations needed for stable training and tests of model calibration. Please see “Interest to the community / key Novelties” in our general response comment, where we elaborate on our contribution.
>
> # Re: Some detailed information in section 3, such as 3.4, would better fit in the experiment section(e.g., section 4.2).
>
> Thank you, we agree and have moved the text to Section 4.2
>
> # Re: What considerations are taken to determine the specific simulation parameters are used in section 4.1? Why is the test set different from the training set and how are the specific parameters determined?
>
> The specific simulation parameters are addressed in our response above.
> The model is trained on electron recoil simulations in the 40-50 keV with Gaussian smearing in the  160 − 466 μm range. For Testing, it makes sense to compare performance in specific cases, i.e. low energy versus high energy and low diffusion versus high diffusion. This way, we test our models in cases where it is harder / easier to determine the direction. Aside from targeting specific test cases, our training and testing data is produced in exactly the same way.

---

> ### Author Response · Authors · 2023-11-28
>
> We've clarified our rationale for prioritizing simulation over real-world evaluation, explained our conservative parameter choices, and demonstrated the model's robustness. Furthermore, we’ve clarified that our contribution is the first probabilistic deep learning framework for predicting 3D directions, along with the necessary approximations for stable training, and tests that empirically demonstrate calibration.
>
> Given the extended discussion period until Dec. 1 and the lower confidence score, we welcome further dialogue. Your feedback is invaluable, and we're eager to address any lingering concerns. If you find our responses satisfactory, we encourage a reconsideration of the rating.

---

### Author Response · Authors · 2023-11-20
**General Response**

We thank the reviewers for their time and valuable feedback. We will address common issues below, and specific questions in the response to each review.

# Comparison to related work

We present a probabilistic DL approach for the task of predicting 3D directions. There is related work from the machine vision community on estimating orientation, which is slightly different. We’ve added the following subsection to better explain the relationship to this work.

“In computer vision, several approaches have applied deep networks to predicting 3D orientation from 2D images (Liao et al., 2019; Huang et al., 2018; Xiang et al., 2018; Mahendran et al., 2017; 2018). Only a handful consider modeling orientation uncertainty (Prokudin et al., 2018; Gilitschenski et al., 2020; Mohlin et al., 2020). All of these approaches are concentrated on predicting orientation. This is similar to predicting direction but different. Orientation includes an additional dimension, which would introduce a spurious degree of freedom for our task.  For example, an airplane’s direction is naturally represented by a point on S2, while the airplane’s orientation requires an additional parameter specifying a rotation about the plane’s main axis. Representing probability distributions over orientations in 3D is non-trivial, and the approaches of Prokudin et al. (2018); Gilitschenski et al. (2020); Mohlin et al. (2020) are all different.

Prokudin et al. (2018) uses Euler angles (pan, tilt, roll) to represent orientation. A unidimensional von Mises distribution over each angle captures orientation uncertainty. However, jointly predicting Euler angles has problems due to degeneracies known as gimbal lock. In Mohlin et al. (2020) orientation is represented by 3D rotation matrices. The orientation uncertainty is estimated by outputting a distribution over SO(3), parameterized by the matrix Fisher distribution. However, this method poorly models classes with rotational symmetries, such as the test case in Appendix B. Gilitschenski et al. (2020) uses unit quaternions as a representation for object orientation. The uncertainty is modeled using the Bingham distribution, an antipodally symmetric distribution over S3. Quaternions do not suffer from gimbal lock and are more compact than rotation matrices. However, the computationally expensive Bingham normalization constant necessitates backpropagation through an interpolator and use of a lookup table.

Our task is not to find the orientation of a rigid body. We want to find the initial direction in the trajectories imaged by our detectors.  The direction lives on S2, so frameworks that output distributions over SO(3) or S3 are inappropriate and fail to leverage a dimension of symmetry. This work proposes an approach focused on predicting directions with distributions on S2. Predicting direction is a general task with many use cases beyond directional recoil detection, including predicting where something is pointing, predicting where someone is looking, determining the source direction of a  signal, etc.”

In the absence of related work on probabilistic direction prediction, we developed two of our own baselines:

1. An algorithm that estimates the best expected directional performance by using truth information that is only accessible in simulation. Previously we named this algorithm Non-ML2 but we have renamed it “Best-Expected” for clarification.
2. A non-probabilistic deep learning model, RCN. This model uses the same architecture but has a deterministic output.
We’ve added a paragraph at the beginning of Section 3 and Section 4.3 to clarify our baselines and their significance.

---

> ### Author Response · Authors · 2023-11-20
> **General Response Continued**
>
> # Interest to the community / key Novelties
>
> We have added the following text to the introduction to provide a better overview of why this contribution is relevant to the ICLR community.
>
> “We develop a deep learning approach to analyze 3D events and probabilistically predict their initial direction. To our knowledge, this is the first deep probabilistic approach for predicting 3D directions with the Von Mises-Fisher distribution on S2. Our framework significantly outperforms traditional algorithms when applied to a simulated electron recoil data set. Leveraging our framework’s uncertainty predictions, performance can be enhanced by discarding high-uncertainty samples. By discarding only the top percentile, our framework achieves the best expected directional performance (as defined in Appendix A). This development is far-reaching as it enhances the sensitivity of all directional recoil detection experiments. Exciting possibilities include aiding in the discovery of dark matter (Billard et al., 2014; O’Hare, 2021)  or resolving the solar abundance problem (Villante & Serenelli, 2019; O’Hare et al., 2022). The framework can also be applied to general problems where direction needs to be predicted, we demonstrate this by applying it to the task of detecting 3D arrows directions in Appendix B.
>
> Our contributions include: 1) The first deep learning framework that predicts direction probabilistically with the von Mises-Fisher distribution on $S^2$, 2) the associated loss function along with approximations that stabilize training, 3) two tests that empirically demonstrate calibration, something which is lacking in related work on predicting orientation, 4) a model capable of analyzing highly-segmented 3D data using state-of-the-art sparse convolution techniques.”
>
> Furthermore, in Section 2.2 we included a detailed discussion positioning our contribution with respect to similar work. We argue that the task of estimating direction, as opposed to estimating orientation, is still an important and general task. Our work is the only deep probabilistic approach focusing on estimating 3D direction.
>
> Finally, we want to emphasize the two tests of calibration that we developed. Similar approaches on estimating orientation with uncertainty often lack in their assessment of model calibration. Prokudin et al. (2018) and Mohlin et al. (2020), do not assess model calibration. Gilitschenski et al. (2020) assess calibration by adding artificial noise to their labels by drawing random perturbations from the Bingham distribution before training. They then demonstrate that the predicted Bingham uncertainty parameters closely match the applied label noise. Our tests assess calibration directly on our test sets without adding noise to labels.
>
> # Real-World Evaluation and the sim-to-real gap
>
> The scope of this paper is to present our framework and compare it against traditional methods and other baselines. In physics, it is common to achieve goals like this using simulation tools such as Degrad. We agree that real-world evaluation is essential for demonstrating the directional performance of a particular detector; however, that is not the goal of this paper. As of now, experimental data that is labeled with the true direction is not readily available. We are actively working on obtaining such data. However, this is an experimentally challenging task that is likely to be presented in a physics venue.
>
> We have added the following text in Section 4.1 to articulate this
>
> “The simulation framework, Degrad, is the standard software package used in physics for simulating the interaction of low-energy electrons in gasses Pfeiffer et al. (2019). The focus of this work is to compare the different models, and our simulations provide a typical setup on which to compare performance. The performance in any particular detector is expected to differ slightly due to the sim-to-real gap. In the future we plan to use radioactive source data to train our models, thereby avoiding the sim-to-real gap entirely.”

---

> > ### Comment · Reviewer_7Dku · 2023-11-20
> > **systematic errors**
> >
> > I mentioned a "sim-real" gap in my review, though I understand that obtaining such data is complicated. Therefore, I encouraged the authors to explore the sources of systematic errors. Even under the assumption that Degrad is a perfect representation of reality, the choice of input parameters for Degrad introduces systematic noise. What happens when you introduce errors in these physical parameters, etc...

---

> > > ### Author Response · Authors · 2023-11-22
> > > **Re: systematic errors**
> > >
> > > Thank you for clarifying.
> > >
> > > This is a list of the parameters that we identify as possible sources of systematic errors.
> > >
> > > 1. ESTART - the starting energy of the electron
> > > 2. FRAC1, FRAC2 - the fraction of the gasses in our gas mixture
> > > 3. T - the temperature
> > > 4. P - the pressure
> > >
> > > The starting energy of the electron strongly impacts the track's directionality. When applying this framework, the model should be trained on the energy range that is being targeted. In our example we train on energies in the range 40-50 keV and test on 40 and 50 keV cases. The starting energy of the recoil is something our detectors directly measure, so this parameter is well-constrained. We also expect the model to generalize slightly beyond the range of energies it is trained on due to natural fluctuations that are captured in the Degrad simulation framework. I.e., in general 52 keV electron tracks are longer and have more ionization than 50 keV tracks; however, it is possible to have a 50 keV track that is longer and has more ionization than a 52 keV track.
> > >
> > > FRAC1, FRAC2, T, P all influence the amount of diffusion in the recoil track. To include the diffusion in our training data, Gaussian smearing is applied to each ionized electron in the track. For each track, the amount of smearing is drawn randomly from a uniform
> > > distribution of 160 − 466 μm. This range is over the minimum and maximum expected diffusion in the setup of Jaegle et al. We then test our trained model on cases that probe specific diffusion values. Because we train our model on a range of diffusion, we expect it to be robust against differences in diffusion caused by slight fluctuations in FRAC1, FRAC2, T, and P.
> > >
> > > FRAC1, FRAC2, T, and P also affect the density of the gas which affects the track shape. FRAC1, FRAC2 can be precisely controlled by purchasing high quality gas mixtures. P can be precisely controlled with a well-calibrated capacitance pressure gauge. We reason that fluctuations in T are the largest source of systematic error that we can identify. In our lab we expected fluctuations in temperature to be less than 2 degrees Celsius. We create a new test set, The counterpart to the 40 keV test set with 443 um diffusion, except simulated at 22 degrees Celsius. By applying our model, which was trained on simulations at 20 degrees celsius, to the 22 degree celsius test set, we can test how robust our framework is to small fluctuations in temperature. The cosine distance loss of the HCN model on the 20 degree test  and 22 degree test sets agrees within 0.48%. We conclude that small fluctuations in temperature have a very small systemic effect on our model.
> > >
> > > It is not a surprise that our model generalizes well here. Small changes in these systematics will cause slight changes in the diffusion and track shape. However, we train our model over a large range of diffusion and there is a significant amount of variability in our electron simulations in terms of track shape and amount of deposited ionization.
> > >
> > > We modified the last paragraph in Section 4.1 to the following
> > >
> > > “The simulation framework, Degrad, is the standard software package used in physics for simulating the interaction of low-energy electrons in gasses Pfeiffer et al. (2019). The focus of this work is to compare the different models, and our simulations provide a typical setup on which to compare performance. The performance in any particular detector is expected to differ slightly due to the sim-to-real gap. In the future we plan to use radioactive source data to train our models, thereby avoiding the sim-to-real gap entirely. Simulation parameters, such as diffusion and binning, were deliberately conservative compared to what is experimentally achievable. We verified that the model performance is robust against modest variation of the Degrad input parameters, such as the gas temperature.”

---

### Author Response · Authors · 2023-11-22
**General comments about sim-to-real gap and systematic errors**

We emphasize that our contribution to the ICLR community is primarily the method for addressing the general problem of probabilistic direction prediction — a contribution whose novelty the reviewers have not disputed. However, the reviewers have asked whether the results can be expected to generalize beyond these simulations to the analysis of experimental data.

In short, the simulated setup was already performance-degraded conservatively, and the model generalizes for different electron and simulation input parameters. The current model is already useful for one major application, which is to guide future detector design work. Hence the generalization to experimental data is of lesser concern at the present stage of our experimental program. Details are provided below.

First, the simulations are performed with very conservative assumptions about the simulation parameters. In past work, detailed studies were performed on comparisons of detector performance versus simulation, and analyses of detector stability [1,2,3].  The detector response to a monoenergetic radioactive source was found to have a very small week-to-week variability (1%), and performance agreed well with simulation. The key detector features identified as reducing performance were the segmentation of the detector readout planes and electron diffusion in the gas, both of which reduce the information available for direction prediction. In our simulations, these two parameters were chosen very conservatively with more information loss than in a standard detector. The simulated detector segmentation is 500 microns, while real-world detectors can have segmentation as small as 50 microns.

As a result of this conservative modeling, systematic effects due to the parameters of the simulation tool (Degrad) become subdominant effects. To explicitly demonstrate this, we have followed the reviewer's suggestion to vary the simulated temperature, and there (see Re: Systematic errors) we show that the variation in performance is tiny. This is expected for physics reasons; for example, the electron diffusion depends on the square root of the temperature (in degrees Kelvin). The range of diffusion already simulated is much larger than the effect of temperature fluctuations. Similar considerations apply to other input parameters and aspects of detector performance.

A second reason why the sim-to-real gap is not a concern is that models like this are being used to refine the detector design, in particular the charge readout electronics. Readout electronics can be simulated with high precision. The goal is to adjust these electronics in simulation to minimize the information loss in the detector. The design goal is for the information loss in the electronics to be smaller than that due to segmentation and diffusion. This will be evaluated by comparing the model performance on different simulated electronics designs in the future. Once the optimized detectors have been constructed, a detailed evaluation of as-built-performance will be made. Then the sim-to-real gap can be explicitly quantified, and the model may be trained or fine-tuned on radioactive source data (i.e. labeled experimental data) to optimize sensitivity.

[1] Vahsen, S. E., et al. "3-D tracking in a miniature time projection chamber." Nuclear Instruments and Methods in Physics Research Section A: Accelerators, Spectrometers, Detectors and Associated Equipment 788 (2015): 95-105.

[2] Lewis, P. M., et al. "Primary track recovery in high-definition gas time projection chambers." The European Physical Journal C 82.4 (2022): 324.

[3] Jaegle, I., et al. "Compact, directional neutron detectors capable of high-resolution nuclear recoil imaging." Nuclear Instruments and Methods in Physics Research Section A: Accelerators, Spectrometers, Detectors and Associated Equipment 945 (2019): 162296.

---

### Meta-Review · Area_Chair_K3DM · 2023-12-11

**Metareview:**

The paper introduces a method for predicting the SO(3) angles of particles from their recoil trajectories within the context of a physics problem. The reviewers are divided in their opinions, expressing doubts about the novelty of the problem and the contribution this research makes to the field of machine learning. A notable point of comparison for this work is Liu et al.'s study, "Delving into Discrete Normalizing Flows on SO(3) Manifold for Probabilistic Rotation Modeling," presented at CVPR 2023. This reference is particularly relevant for the task of SO(3) prediction with uncertainty. Given the concerns raised by the reviewers, particularly regarding the need for a more thorough exploration of the novelty and contribution of the research, the area chair is currently leaning towards rejecting the submission. It is suggested that the authors comprehensively address these main concerns before resubmitting to another major machine learning conference/journal.

**Justification For Why Not Higher Score:**

Given the concerns raised by the reviewers, particularly regarding the need for a more thorough exploration of the novelty and contribution of the research, the area chair is currently leaning towards rejecting the submission.

**Justification For Why Not Lower Score:**

N/A

---

### Decision · Program_Chairs · 2024-01-16

Reject